# β-Cyclodextrin Inclusion Complex Containing *Litsea cubeba* Essential Oil: Preparation, Optimization, Physicochemical, and Antifungal Characterization

**Yinhong Wang** [1,2] , **Chunxiao Yin** [1,2] **, Xiaomei Cheng** [1,2] **, Gaoyang Li** [2] **, Yang Shan** [2,*] **and Xiangrong Zhu** [1,2,*]

[1] Longping Branch Graduate School, Hunan University, Changsha 410125, China; nostalgia99yhw@163.com (Y.W.); chunxiaoyin@scbg.ac.cn (C.Y.); xiaomei2020@hnu.edu.cn (X.C.)

[2] Hunan Provincial Key Laboratory for Fruits and Vegetables Storage Processing and Quality Safety, Hunan Agricultural Product Processing Institute, Hunan Academy of Agricultural Sciences, Changsha 410125, China; lgy7102@163.com

\* Correspondence: shanyang_jgs@163.com (Y.S.); xiangrongchu@163.com (X.Z.); Tel.: +86-731-8469-1289 (Y.S.); +86-731-8469-1006 (X.Z.)

**Abstract:** *Litsea cubeba* essential oil (LCEO), as naturally plant-derived products, possess good antimicrobial activities against many pathogens, but their high volatility and poor water solubility limit greatly the application in food industry. In this research, inclusion complex based on β-cyclodextrin (β-CD) and LCEO, was prepared by saturated aqueous solution method. An optimum condition using the response surface methodology (RSM) based on Box–Behnken design (BBD) was obtained with the inclusion time of 2 h and β-CD/LCEO ratio of 4.2 at 44 °C. Under the condition, the greatest yield of 71.71% with entrapment efficiency of 33.60% and loading capacity of 9.07% was achieved. In addition, the structure and characteristic of LCEO/β-CD inclusion complex (LCEO/βCD-IC) were investigated using scanning electron microscopy (SEM), X-ray diffraction (XRD), and Fourier transform infrared spectroscopy (FT-IR), which indicated that LCEO/βCD-IC was successfully formed. The particle size of LCEO/βCD-IC was determined to be 17.852 μm. Thermal properties of LCEO/βCD-IC evaluated by thermogravimetric-differential scanning calorimetry (TG-DTA) illustrated better thermal stability of the aimed product compared with the physical mixture. Furthermore, the tests of antifungal activity showed that LCEO/βCD-IC was able to control the growth of *Penicillium italicum*, *Penicillium digitatum,* and *Geotrichum citri-aurantii* isolated from postharvest citrus. Our present study confirmed that LCEO/βCD-IC might be further applied as an alternative to chemical fungicides for protecting citrus fruit from postharvest disease.

**Keywords:** *Litsea cubeba* essential oil; inclusion complex; response surface methodology; antifungal activity; postharvest disease

## 1. Introduction

Nowadays, interests toward the application of natural antimicrobial agents for food preservation have motivated the study of essential oils (EOs) [1,2]. *Litsea cubeba*, popularly known as an aromatic plant, belongs to the *Lauraceae* family, *Laurales* [3]. Its fruit is pepper-like and the trees are widely distributed in Southern China, Indonesia, and other countries of Southeast Asia [4,5]. As one of the important economic spice tree species in China [6], *Litsea cubeba* fruit can be used as herbal medicine for gastrointestinal discomfort, respiratory diseases, and inflammation treatment [7,8]. *Litsea cubeba*

essential oil (LCEO), a pale yellow liquid with strong aroma, has been categorized as generally recognized as safe (GRAS) by the Food and Drug Administration (FDA) [9]. LCEO has been extensively applied as edible food additives or cosmetic flavoring agents [10], and it is also a raw material of extracting citral, vitamin A, E, K, and ionone [11]. Moreover, LCEO possesses good antibacterial activities against a broad range of foodborne pathogens [12–14]. However, applying LCEO for food preservatives is limited because of their poor water solubility, high volatility in natural ambient temperature conditions, and sensitivity to oxidative decomposition [15,16].

β-cyclodextrin (β-CD), belongs to the family member of cyclic oligosaccharides [17], which has a central hydrophobic inner cavity and an outward hydrophilic surface, and it has strong complexing capacity towards small molecules [18]. β-CD commonly served as a microcapsule coating material [19]. Numerous organic guest molecules like EOs can be entrapped into the center cavity of β-CD to form an inclusion complex without altering the chemical structure of β-CD [20]. The aqueous solubility and physical and chemical stability of guest molecules could be strengthened after complexation with β-CD [21], and the molecular volatility can be weakened [22]. From the technological point of view, the complexation of EOs with CDs has some additional advantages, such as improvement of handling of dry powers, mask of undesirable odors, and reduction of storage costs. β-CD-based LCEO inclusion complex could control the LCEO release at a certain temperature and suitable humidity [23], which is beneficial to the LCEO availability and efficiency utilized in the food industry. Figure 1 presented the forming scheme of LCEO/ β-CD inclusion complex.

To improve the utilization rate of LCEO, the optimal parameters of complexation process have to be investigated carefully [24]. Traditional experimental designs for optimization is costly and time-consuming [25]. However, response surface methodology (RSM) is a numerical and statistical means that applies quadratic polynomial models to determine the exact relationship between one or more response variables and independent variables [26,27], it is a useful technique that can model, analyze, and optimize the experimental responses influenced by several factors [28]. Box–Behnken design (BBD) is a multivariate optimization approach based on three-level factor design in the RSM, which could build the second-order response surface models [24,29]. It is better than other response surface designs due to lower costs of carrying out experimentation and more valid experimental design [30,31]. Many researches have indicated that RSM is a helpful technique to optimize conditions for the encapsulation formulation [32–34], and the principal advantage of employing RSM is that it demands few experimental runs to establish the dependence between variable parameters and response values [35,36]. Thus, our work mainly aimed to assess the effects of β-CD/LCEO ratios, preparation temperature, and time on the yield of inclusion complex by applying RSM based on BBD. Therefore, the technological optimum condition for preparing LCEO/β-CD inclusion complex (LCEO/βCD-IC) was ascertained. Then the characterization of physicochemical properties of LCEO/βCD-IC was evaluated to further confirm the successful entrapment of LCEO into the cavity of β-CD.

In previous studies, antimicrobial activities of LCEO have been extensively investigated. However, most of them concentrated on the assays against bacterial pathogens [14,15,37], few studies on the effectiveness of free LCEO and LCEO/β-CD inclusion complex against fungal pathogens have been reported to date [38], especially those fungi infecting fruits and vegetables. Citrus is one of the most economically important fruit crops in the world, which is rich in vitamin C, flavonoids, carotene, iron, and other nutrients [39]. However, they are susceptible to infection by fungal pathogens during storage, which can lead to serious decay and significant economic losses [40,41]. Blue mold, green mold, and sour sot, which are respectively caused by filiform fungi *Penicillium italicum*, *Penicillium digitatum*, and *Geotrichum citri-aurantii*, which are the most common fungal diseases infecting citrus fruit worldwide [42]. So, in our present study, the antifungal property of the samples was preliminarily investigated on these fungi, which can contribute to the development of green preservatives for food and provide an alternative to chemical and toxic fungicides.

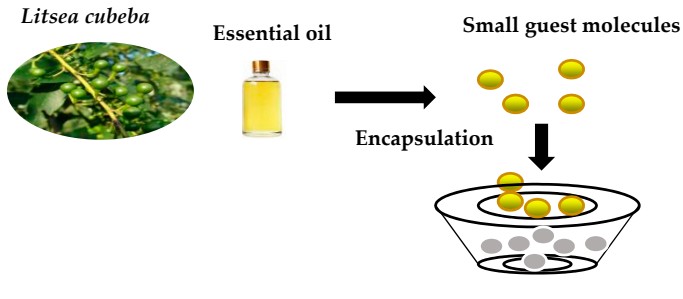

**Figure 1.** The forming scheme of *Litsea cubeba* essential oil (LCEO)/β-cyclodextrin (β-CD) inclusion complex.

## 2. Materials and Methods

### 2.1. Materials

*Litsea cubeca* essential oil was purchased from the FeiBo Biotechnology Co., Ltd. (Guangzhou, China) with 99% purity. β-cyclodextrin was purchased from Sinopharm Chemical Reagent Co., Ltd. (Shanghai, China). Absolute ethanol (AR grade) was provided by KeLong Chemicals Co., Ltd. (Chengdu, China).

### 2.2. Pathogens

The molds of *Penicillium italicum*, *Penicillium digitatum*, and *Geotrichum citri-aurantii* used in this study were isolated from infected citrus fruit with clear mold symptoms, provided by South China Botanical Garden, Chinese Academy of Sciences. The fungi were preserved on potato dextrose agar (PDA) at 28 ± 2 °C.

### 2.3. Preparation of the Inclusion Complex and the Physical Mixture

The LCEO/βCD-IC was prepared using saturated aqueous solution method with slight modifications [21,43]. Firstly, a certain amount of β-CD was dissolved in distilled water at 40 °C under continuous stirring in a thermostatic magnetic agitator to make β-CD dissolve completely, and transparent solution was obtained. Then, a solution of LCEO in ethanol (1:10, w/v) was added slowly to the β-CD solution. Subsequently, the mixture was stirred continuously for a definite time, then cooled down to the room temperature and placed at 4 °C overnight. The mixture was filtered, washed twice with double distilled water and absolute ethanol to wash away the excess β-CD and LCEO. Finally, the solid powder was collected and dried in 50 °C drying oven until a constant dry weight was achieved. The physical mixture was prepared by the reported method [44]. Certain weight of β-CD and LCEO were mixed and ground in a mortar until a homogeneous physical mixture. They were stored in a hermetically sealed glass jar at room temperature for further use.

### 2.4. Experimental Design

#### 2.4.1. Single-Factor Tests

In this section, three independent variables (β-CD/LCEO ratios, preparation temperature, and time) were selected for the single-factor experiments. The ranges of tested variables were as follows: β-CD/LCEO ratios of 1, 2, 4, 6, 8, and 10; preparation temperatures of 20, 30, 40, 50, 60, and 70 °C; and preparation times of 1.0, 1.5, 2.0, 2.5, and 3.0 h. The effects of these variables on the yield of LCEO/βCD-IC was studied. The yield was calculated by the following equation.

$$Y(\%) = W_M/(W_1 + W_2) \times 100\% \tag{1}$$

where $W_M$ is the weight of inclusion complex and $W_1$ is the initial added amount of LCEO. $W_2$ is the added amount of β-CD.

### 2.4.2. Optimization of LCEO/βCD-IC Preparative Conditions

Response surface methodology was used to determine the optimum condition of inclusion process. Based on the single-factor experiment above, the effects of three independent variables: The ratio of β-CD/LCEO ($X_1$: 2–6 *w/w*), the temperature ($X_2$: 30–50 °C), and preparation time ($X_3$: 1.5–2.5 h) on the response variable: Inclusion yield (Y), were evaluated using RSM with BBD.

The three independent variables were run at three levels in terms of BBD for three coded levels (−1, 0, and 1), factors and their levels used in the experimental design are shown in Table 1. All experiments were carried out in triplicate. The BBD is performed to investigate the influences of each independent variable and the interaction between independent variables on the response. These effects could be evaluated using the second-order polynomial regression equation.

$$Y_n = a_0 + a_{1 \times 1} + a_2 X_2 + a_3 X_3 + a_{12} X_1 X_2 + a_{13} X_1 X_3 + a_{23} X_2 X_3 + a_{11} X_1^2 + a_{22} X_2^2 + a_{33} X_3^2 \quad (2)$$

where $Y_n$ represents response function, $a_0$ is a constant, $a_1$, $a_2$, and $a_3$ are the coefficients of the linear effects, $a_{12}$, $a_{13}$, and $a_{23}$ are the coefficients of the interaction between the factors, and $a_{11}$, $a_{22}$, and $a_{33}$ are the coefficients of the quadratic effect.

**Table 1.** Actual levels at coded factor levels of independent variables applied in the response surface methodology (RSM).

| Symbol | Independent Variables | Actual Levels at Coded Factor Levels | | |
|---|---|---|---|---|
| | | −1 | 0 | 1 |
| $X_1$ | Ratio of β-CD: LCEO (g/g) | 2:1 | 4:1 | 6:1 |
| $X_2$ | Temperature (°C) | 30 | 40 | 50 |
| $X_3$ | Preparation time (h) | 1.5 | 2 | 2.5 |

### 2.5. Characterization of LCEO/βCD-IC

### 2.5.1. Determination of Entrapment Efficiency (EE%) and Loading Capacity (LC%)

EE% and LC% are two important parameters evaluating the coverage percentage of guest molecules embedded within host material. The LCEO sample was diluted with absolute ethanol to form an exact concentration, and the solution was scanned over the range of 200–400 nm using an ultraviolet visible spectrophotometer, the maximal absorption was observed at 262 nm. Then the absorbance values at this wavelength for different concentrations of the ethanol-diluted LCEO ranged from 1.6 to 3.2 ($10^{-4}$ g mL$^{-1}$) were recorded, and the calibration curve was plotted.

To determine the LCEO content in LCEO/βCD-IC, 100 mg of the IC was placed in mortar to be ground fully, then the powder was dissolved in ethanol and washed to filter the residues, and the remaining solution was diluted with ethanol again to get a final volume of 10 mL. The absorbance of the LCEO solution was measured at 262 nm and its unknown concentration was calculated based on the calibration curve of LCEO. All measurements were conducted for three replications. The indexes were measured through the following equations [23].

$$EE\% = W_0/W_1 \times 100\% \quad (3)$$

$$LC\% = W_0/W_M \times 100\% \quad (4)$$

where $W_0$ is the total amount of loaded LCEO.

### 2.5.2. Fourier Transform Infrared Spectroscopy (FT-IR)

FT-IR spectra of LCEO, β-CD, physical mixture, and LCEO/βCD-IC were recorded using a FT-IR spectrometer (Bruker Alpha, Germany) with a wavenumber range of 400–4000 cm$^{-1}$, and 64 scans were collected for each sample. A drop of LCEO was added to the surface of a potassium bromide (KBr) disc. As for other three solid samples, different samples were ground respectively with KBr reagent and compressed into small tablets under a pressure of 12 MPa.

### 2.5.3. X-ray Diffraction (XRD)

The XRD patterns of β-CD, physical mixture and LCEO/βCD-IC were performed using a D/max9 diffractometer with Cu Kα radiation (40 kV, 30 mA) at a scanning rate of 5°/min, these powder samples were mounted on a hyaloid sample holder and then scanned in a 2 theta range of 3–60 at 0.02°/s. The scanning mode was continuous.

### 2.5.4. Determination of Particle Size

The size distributions of β-CD, physical mixture and LCEO/βCD-IC were measured using a laser light scattering particle sizer (Mastersizer-2000) (Malvern Panalytical Ltd., Malvern, UK) and certain volume of distilled water as a suspending solution. The area-volume mean diameter, D [4, 3], was recorded.

### 2.5.5. Scanning Electron Microscopy (SEM)

The surface morphological structure of different samples including β-CD, physical mixture, and LCEO/βCD-IC were observed by EVO-LS10 scanning electron microscopy (ZEISS, Shanghai, China), operated at 10.00 kV at different levels of magnification. The samples were fixed on SEM stubs using double-sided adhesive. Then these samples were coated with gold-palladium electroplating (30 s, 1.8 mA, 2.4 kV) in a SEM Coating System sputter coater before scanning to ensure complete conductivity.

### 2.5.6. Thermogravimetric Analysis (TGA)

The thermal behaviors of three samples were measured by a thermal analyzer. Approximately 5 mg of the samples were placed in crucible each time. Then, 99.99% Nitrogen was employed as a dynamic atmosphere at 100 mL/h. Heating rate was 20 °C/min from 30 to 500 °C.

### 2.5.7. Vitro Antifungal Activity Assays

Determination of Minimum Inhibitory Concentration (MIC) and Minimum Bactericidal Concentration (MBC)

MIC for LCEO was determined by microdilution broth method in sterile 96 well microplates [45]. Briefly, LCEO was dissolved with absolute ethyl alcohol solution. Aliquots of 100 μL PDB broth were added to the microplate wells. Then 100 μL of LCEO solution (400 mg mL$^{-1}$) was added to the first well of the row, and the targeted concentrations were achieved by serial two-fold method. The concentrations of LCEO in wells for the three fungi were: 100, 50, 25, 12.5, 6.25, 3.13, 1.56, 0.78, 0.39, 0.195, 0.098, and 0.049 mg mL$^{-1}$. Aliquots of 100 μL of all microbial inoculums with adjusted concentration of $10^6$–$10^7$ CFU/mL were transferred to the wells, the microplates were covered with sealing film to prevent LCEO solution from evaporating, and incubated at 28 °C for 48 h. The MICs were defined as the lowest concentration where no visible microbial growth observed in wells after 48 h of incubation. For MBC determination, aliquots of 100 μL from suspensions of the tested groups without visible fungi growth were subcultured on PDA agar for 48 h at 28 °C, and the lowest concentration of LCEO that caused the complete absence of colonies growth was defined as MBC.

Preliminary Antifungal Activity Assessment of LCEO/βCD-IC

Agar disc diffusion method was employed to assess the antifungal activity preliminarily [46]. Samples of 0.5 g β-CD and LCEO/β-CD-IC were individually dispersed in 20 mL of PDA medium, and the obtained mixture was immediately poured into sterilized Petri dishes (90 mm in diameter). After solidifying, a 6 mm diameter disc of inoculum, taken from the edge of 7-day-old cultures of fungi, was respectively transferred to the center of each Petri dish. PDA dishes adding nothing as blank control. After incubation for 48 h at 28 ± 2 °C, the colony diameter was measured with a digital caliber [47]. Each treatment was performed in triplicate.

*2.6. Statistical Analysis*

All data were expressed as mean ± standard deviation (SD) for triplicate experiments, and the analysis of variance (ANOVA) was used to analyze the statistically significant differences between the means of independent variables using IBM SPSS Statistics Software (Version 25.0, IBM SPSS Inc., Armonk, NY, USA). A stepwise procedure was employed to create the three-dimensional surface plots for investigating the relationship between a response and three independent variables using Design-Expert Software (version 8.0.0, Stat-Ease, Inc., Minneapolis, MN, USA). Differences among samples were analyzed using Duncan's multiple range test ($p < 0.05$).

## 3. Results

*3.1. Single-Factor Test*

Yield of LCEO/βCD-IC was influenced by several factors, such as β-CD/LCEO ratios, preparation temperature, and time. As shown in Figure 2, Yield of LCEO/βCD-IC maximized when β-CD/LCEO ratio reached 4:1. When temperature ranged between 20 and 40 °C. The yield was increasingly improved, especially attained the greatest yield at 40 °C. The inclusion yield was gradually increased when the preparation time was from 1.0 to 2.5 h.

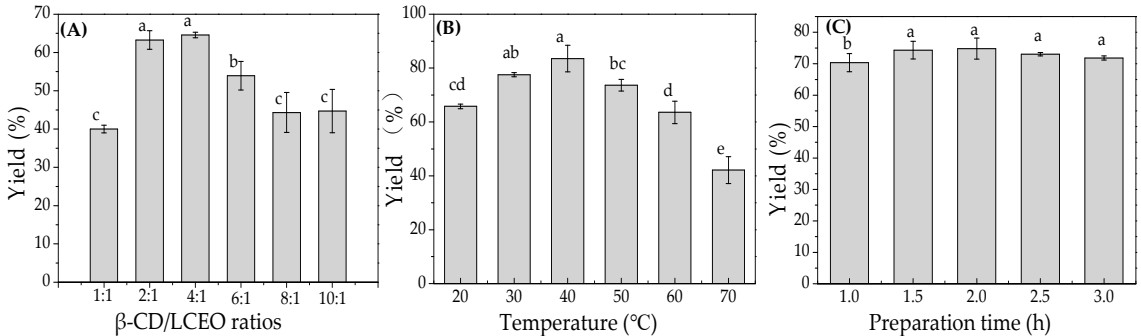

**Figure 2.** Effects of β-CD/LCEO ratios (**A**), temperature (**B**), and preparation time (**C**) on yield of LCEO/β-CD inclusion complex.

*3.2. Optimization of LCEO/βCD-IC Preparative Conditions Using RSM*

3.2.1. Establishment of the Regression Model

To optimize the three parameters for preparing LCEO/βCD-IC, the three-factor BBD was applied based on 3 coded levels of 3 independent variables, leading to 17 assays with 5 replicates at the center of the experimental design.

Then, 17 formulations of the LCEO/βCD-IC with 3 factors were designed and the produced yield were shown in Table 2, quadratic polynomial regression analysis was performed on the experimental

data by using Design-Expert 8.0.0 software, the regression equation describing the mathematical relationships between the yield and the selected factors was constructed (Equation (5)).

$$\text{Yield}(\%) = 70.56 + 0.68X_1 + 1.59X_2 - 0.79X_3 + 2.32X_1X_2 - 0.13X_1X_3 - 1.60X_2X_3 - 11.05X_1^2 - 2.50X_2^2 - 3.26X_3^2 \quad (5)$$

**Table 2.** Box–Behnken design with independent and response variables for preparation of LCEO/β-CD inclusion complex in different conditions.

| Run No. | Factors | | | Response Variable |
|---|---|---|---|---|
| | Ratio of β-CD: LCEO ($X_1$) | Inclusion Temperature ($X_2$) | Inclusion Time ($X_3$) | Yield (%) |
| 1 | 6 | 50 | 2 | 60.00 |
| 2 | 2 | 40 | 1.5 | 56.22 |
| 3 | 4 | 40 | 2 | 71.07 |
| 4 | 6 | 40 | 1.5 | 59.43 |
| 5 | 4 | 30 | 1.5 | 60.80 |
| 6 | 2 | 40 | 2.5 | 53.33 |
| 7 | 2 | 30 | 2 | 58.67 |
| 8 | 4 | 40 | 2 | 72.80 |
| 9 | 2 | 50 | 2 | 55.56 |
| 10 | 4 | 40 | 2 | 69.33 |
| 11 | 4 | 40 | 2 | 69.60 |
| 12 | 6 | 40 | 2.5 | 56.00 |
| 13 | 4 | 40 | 2 | 70.00 |
| 14 | 4 | 30 | 2.5 | 64.00 |
| 15 | 6 | 30 | 2 | 53.81 |
| 16 | 4 | 50 | 1.5 | 68.80 |
| 17 | 4 | 50 | 2.5 | 65.60 |

Table 3 presents the coefficient of multiple determinations ($R^2$) and the adjusted coefficient of multiple determinations (adjusted $R^2$) of inclusion yield, indicating whether a regression model is adequate. ANOVA was used to evaluate the significance of the coefficient of the quadratic polynomial equation. The results showed that the model is significant ($p < 0.01$), indicating that the quadratic equation model is extremely significant, the lack of fit test ($p > 0.05$) was not significant on inclusion yield, implying that the experimental error was minor, and the model equation fitted the results well, which could replace the real point of the experiment to analyze and predict the response value. In addition, the $R^2$ value reached up to 0.9670, indicating that the regression model was quite efficient for fitting the data under the condition of test, the adjusted $R^2$ value was 0.9247, demonstrating that a favorable agreement between the predicted and the experimental values of the model for the yield of LCEO/βCD-IC. The value of precision of this model was 11.992, which was more than 4, revealing that the model was feasible.

The independent variables with largest to smallest effects on the yield were the quadratic effects of β-CD/LCEO ratio and preparation time, respectively ($p < 0.01$), followed by the linear effect of temperature, the interaction effect of ratio and temperature, and the quadratic effect of temperature ($p < 0.05$). The two linear effects of ratio and time and the other two interaction effects had no impact on yield. F-value could also reveal the effects of factors on yield, the larger the value was, the stronger the influence on the yield was. The F-value of temperature was the largest, indicating that such factor had the highest effect on the response, followed by preparation time and β-CD/LCEO ratios. Therefore, the model of RSM with BBD was suitable for optimizing technical parameters of preparation of LCEO/βCD-IC.

**Table 3.** Analysis of variance (ANOVA) of the regression coefficients of the fitted quadratic equations for inclusion yield (Y).

| Independent Variable | Inclusion Yield(Y) | | | | | |
|---|---|---|---|---|---|---|
| | Sum of Squares | df | Mean Square | F-Value | *p*-Value | Significance |
| Model | 683.42 | 9 | 75.94 | 22.82 | 0.0002 | significant |
| Linear effect | - | - | - | - | - | - |
| $X_1$ | 3.73 | 1 | 3.73 | 1.12 | 0.3251 | - |
| $X_2$ | 20.10 | 1 | 20.10 | 6.04 | 0.0436 | * |
| $X_3$ | 4.99 | 1 | 4.99 | 1.50 | 0.2602 | |
| Interaction effect | - | - | - | - | - | - |
| $X_IX_2$ | 21.62 | 1 | 21.62 | 6.50 | 0.0382 | * |
| $X_1X_3$ | 0.073 | 1 | 0.073 | 0.022 | 0.8865 | - |
| $X_2X_3$ | 10.24 | 1 | 10.24 | 3.08 | 0.1228 | - |
| Quadratic effect | - | - | - | - | - | - |
| $X_1{}^2$ | 514.35 | 1 | 514.35 | 154.57 | <0.0001 | ** |
| $X_2{}^2$ | 26.26 | 1 | 26.26 | 7.89 | 0.0262 | * |
| $X_3{}^2$ | 44.82 | 1 | 44.82 | 13.47 | 0.0080 | ** |
| Residual | 23.29 | 7 | 3.33 | - | - | - |
| Lack of Fit | 15.27 | 3 | 5.09 | 2.54 | 0.1951 | not significant |
| Pure error | 8.03 | 4 | 2.01 | - | - | - |
| Cor total | 706.71 | 16 | - | - | - | - |
| $R^2$ | 0.9670 | - | - | - | - | - |
| Adjusted $R^2$ | 0.9247 | - | - | - | - | - |
| Adeq precision | 11.992 | - | - | - | - | - |

Note: * means that the difference is significant; ** means that the difference is extremely significant.

Experimental data were analyzed further. The results suggested that the quadratic polynomial equation was able to illustrate the response surface plots and predict the yield of LCEO/βCD-IC. Figure 3a–c are the contour plots of two factors of interaction effect. The contour plot of interaction between β-CD/LCEO ratios and preparation time looked like circle (Figure 3a), implying that their interaction effect was not significant. Furthermore, the plots of interactions between temperature and time, ratio, and temperature were elliptic, indicating that they had very significant interaction effects.

Figure 3d–f presents the plots of three-dimensional response for yield of LCEO/βCD-IC. The yield was smallest when the inclusion time was 1.5 h. With time extended, it was slowly increased but gradually decreased when time was more than 1.9 h. At constant preparation time, increasing β-CD/LCEO ratios could result in better yield, but the yield began to decline when the ratio was about 4.5 (Figure 3d). However, the yield firstly improved rapidly with temperature increasing, then tended to be stable when the temperature was close to 45 °C (Figure 3e). This was probably because LCEO could not to go into the cavity of β-CD at lower temperature.

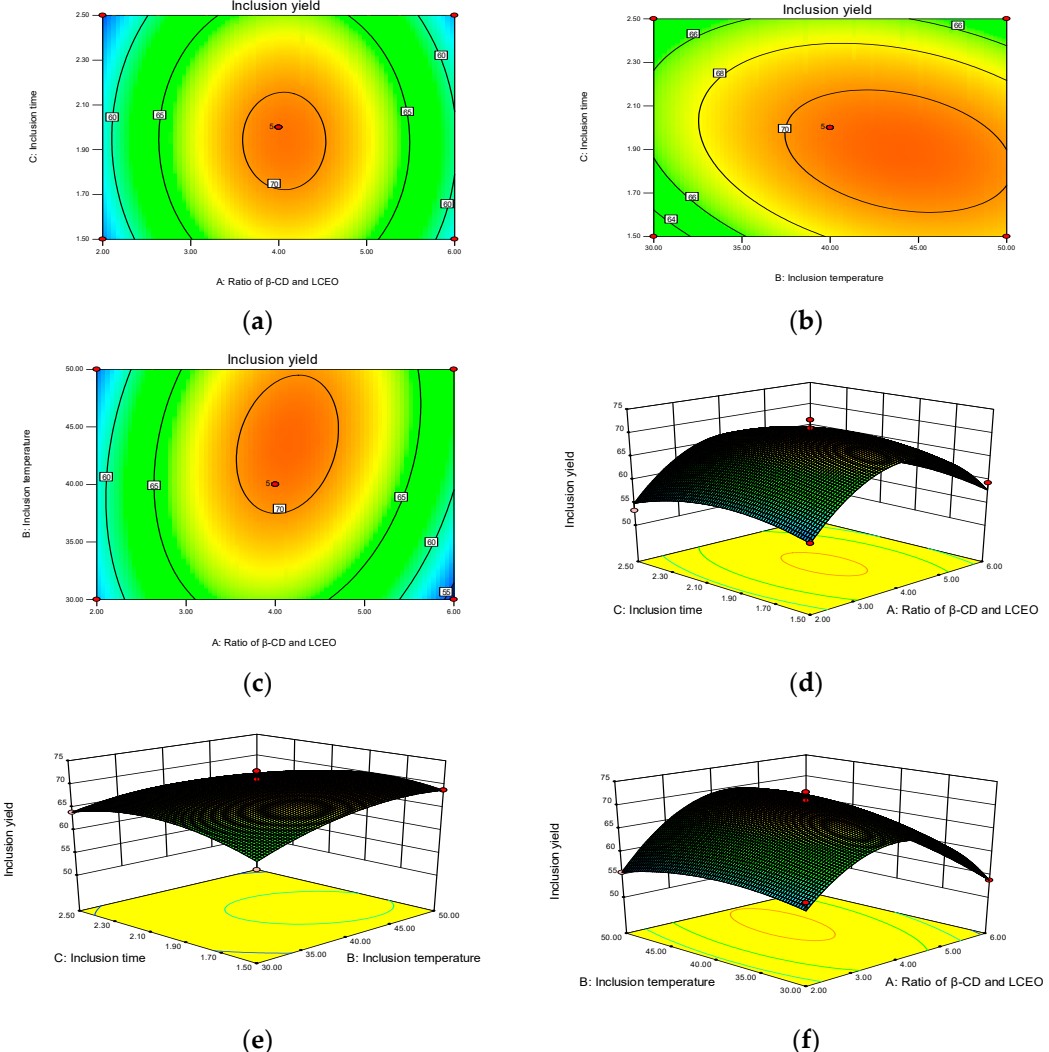

**Figure 3.** Contour plots (**a**–**c**) and response surfaces (**d**–**f**) showing effects of the interaction between ratio of β-CD and LCEO and inclusion time (**a**,**d**), inclusion time and temperature (**b**,**e**), and ratio of β-CD and LCEO and inclusion temperature (**c**,**e**) on inclusion yield.

### 3.2.2. Verification Test

The response surface graph of three factors predicting the yield of LCEO/βCD-IC according to regression equation directly reflected the effect of every factor on response value. At the selected range of factors, the theoretical and optimum condition of β-CD/LCEO ratio of 4.16, temperature of 44.27 °C, and time of 1.89 h was obtained, which provided optimal yield of 71.01%. The condition of parameters were adjusted to a ratio of 4.2, temperature of 44 °C, and time of 2 h considering the practical operability. Then the verification experiment was carried out at this technological condition, the inclusion yield of 71.71% was obtained, whose standard deviation was only 0.58, such a result was close to the predicted value, indicating that the regression equation could fit the actual situation well and the model was correct and even reliable.

### 3.3. EE% and LC%

EE% is a quantitative parameter to quantify the amount of active compound entrapped in the inclusion complex. Value of LC% can be used to calculate the theoretical oil loading of the inclusion complex. In the present study, EE% and LC% of LCEO/βCD-IC was 33.60 ± 0.08% and 9.07 ± 0.02%, respectively. It is worth noting that the difference of EE% and LC% was correlated with the EO variety,

EO concentration, as well as the complexation temperature and time, however, the drying method applied in preparation of inclusion complex also mattered. LCEO could be released from the IC in the aqueous circumstances, which might be influenced by the EE%, thus affected the antimicrobial activity of LCEO/βCD-IC, as reported by a previous study [48].

### 3.4. FT-IR Analysis

FT-IR technique is commonly used to confirm the presence of guest molecule in host materials and the interaction between them in solid phase [49]. The FT-IR spectrum of LCEO, β-CD, physical mixture, and the β-CD/LCEO-IC were displayed in Figure 4. In the β-CD infrared spectrum (Figure 4a), there were several and obvious characteristic peaks of absorption. The peak at 3385.733 cm$^{-1}$ represents the stretching vibration of –OH group in β-CD. While the band detected at 2923.855 cm$^{-1}$ was related to the vibration of stretching of –CH$_3$ group and the peak at 1643.38 cm$^{-1}$ was the result of H–O–H bending vibration. The absorption peak near at 1158.82 cm$^{-1}$ and 1028.917 cm$^{-1}$ resulted from the stretching vibration of C–O and C–O–C groups in the inner cavity of β-CD, respectively. Finally, the peaks at 756.7382 cm$^{-1}$ represents the in-plane rocking vibration of –CH$_2$ [37].

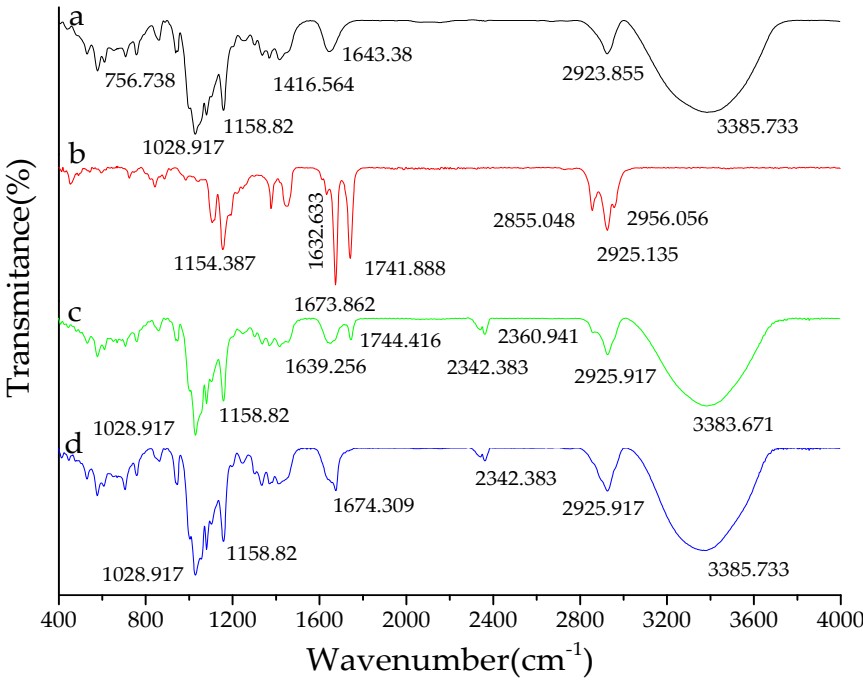

**Figure 4.** FT-IR spectra of β-CD (**a**), LCEO (**b**), the physical mixture (**c**), and the LCEO/β-CD inclusion complex (LCEO/βCD-IC) (**d**).

In the infrared spectrum of LCEO (Figure 4b), the typical strong absorption band monitored at 1673.862 cm$^{-1}$ was associated with the stretching vibration of C=O group in citral compound. Meanwhile, the peak that denoted C=C vibration of stretching was detected at 1632.633 cm$^{-1}$, the peaks at 2956.056, 2925.135, and 2855.048 cm$^{-1}$ represent the vibration of –CH$_3$ and –CH$_2$, respectively. For the spectrum of physical mixture (Figure 4c), its main peaks appeared at 3383.671 cm$^{-1}$ (for O–H stretching vibration), 2925.917 cm$^{-1}$ (for –CH$_2$ stretching vibration), and 1639.256 cm$^{-1}$ (for H–O–H stretching vibration), which was almost exactly similar to β-CD, it owned the common peaks of LCEO and β-CD, the absorption peak of 1741.888 cm$^{-1}$ in LCEO shifted to another wave number of 1744.416 cm$^{-1}$.

Nevertheless, some effective distinction between LCEO and the LCEO/βCD-IC were revealed in the FT-IR spectrum of IC (Figure 4d). The prominent peak of stretching mode of C=O functional group was shifted from 1673.862 cm$^{-1}$ to 1674.309 cm$^{-1}$, and the peak intensity decreased obviously,

indicating that LCEO was included into the cavity of β-CD, the stretching vibration of C=O was limited and it was fixed in the lipophilic cavity of β-CD. The hydrogen intermolecular bond between LCEO and β-CD was formed, as well as the hydrophobic force of van der Waals and electrostatic forces. In addition, the characteristic absorption peaks of LCEO at 2855.048 cm$^{-1}$, 2956.056 cm$^{-1}$, and 1632.633 cm$^{-1}$ disappeared from the spectrum of the IC, indicating a strong physical cross-linking between LCEO and β-CD. Based on these conclusions, it could be initially inferred that LCEO entered the inner cavity of β-CD molecule.

### 3.5. XRD Analysis

The powder XRD patterns of β-CD, physical mixture and LCEO/βCD-IC were shown in Figure 5. As illustrated in Figure 5A, the β-CD was in a crystalline form, various diffraction peaks were observed at different angles (2θ) of 4.52°, 9.00°, 10.66°, 12.64°, 18.82°, 22.70°, 27.06°, 31.98°, 34.78°, 39.64°, 44.06°, 51.36°, and so on. Figure 5B presents the X-ray powder diffraction spectrum of the physical mixture, which was not much different from the XRD pattern of β-CD, indicating that no new phase was formed. However, when the LCEO molecule was included in the cavity of β-CD, as shown in Figure 5C, most of the crystalline diffraction peaks of β-CD disappeared after complexation with LCEO, such as the characteristic diffraction peaks at 2θ value of 12.64°, 22.70°, and 27.06° of β-CD in inclusion complex were disappeared, while the typical peaks at 2θ value of 5.78°, 6.50°, 11.60°, 18.00°, 20.82°, and 23.56° appeared expectably, indicating that β-CD molecules were reoriented after LCEO was located in the β-CD cavity, and a new crystal type that significantly differed from β-CD and the physical mixture was formed. Moreover, the molecular organization of LCEO/βCD-IC also transformed after complexation.

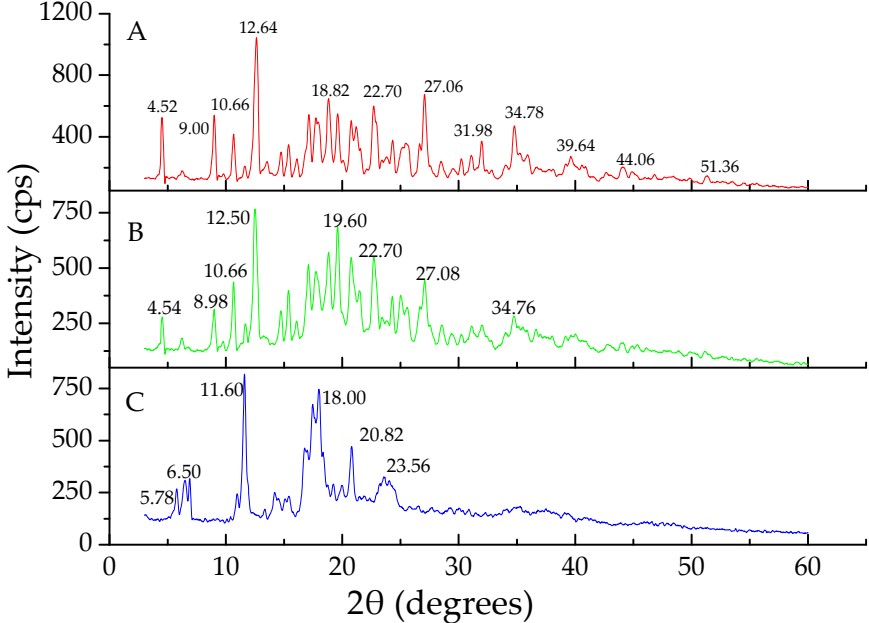

**Figure 5.** X-ray diffraction patterns of β-CD (**A**), physical mixture (**B**), and the LCEO/β-CD inclusion complex (**C**).

### 3.6. Particle Size Analysis

As depicted in Figure 6, the particle sizes distribution of β-CD and LCEO/βCD-IC was relatively uniform. The particle size of β-CD was mainly distributed in the range of 80–100 μm, the area-volume mean diameter (D [4, 3]) obtained was 94.858 μm. However, the powder size became more smaller after encapsulation, the size of which principally concentrated in the range of 5–20 μm, its D [4, 3] was about 17.852 μm. The particle size distribution of physical mixture was irregular and abnormal.

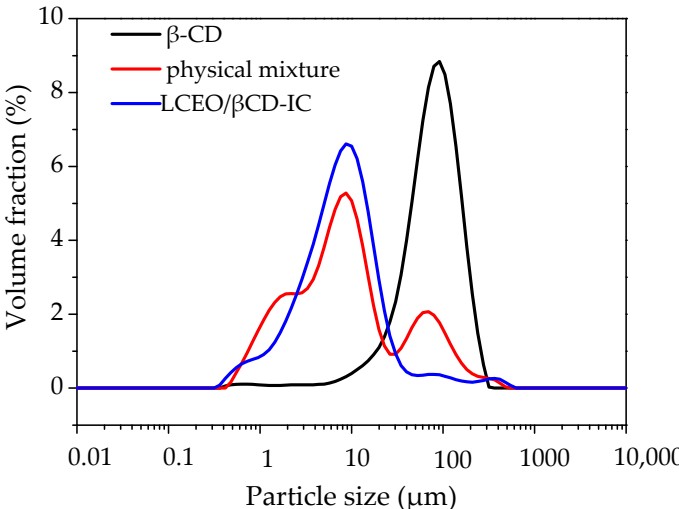

**Figure 6.** Distribution of particle sizes of β-CD, physical mixture and the LCEO/β-CD inclusion complex.

### 3.7. SEM Analysis

SEM can be used to further confirm the surface morphology of β-CD, physical mixture, and the optimal LCEO/βCD-IC prepared by RSM experiment. Results of images presented that β-CD (Figure 7A) was blocky and irregular crystal shapes with large particles, and its structure was relatively dense with strong integrity. As for the physical mixture, a milky and opaque appearance was observed on the surface of β-CD (Figure 7B), speculating that LCEO only adhered onto the exterior surface of β-CD and failed to be included into the cavity of β-CD. For LCEO/βCD-IC, smaller crystal particles with well-organized shapes could be viewed (Figure 7C), and the particle size tended to be uniform, which clearly exhibited a stronger aggregation.

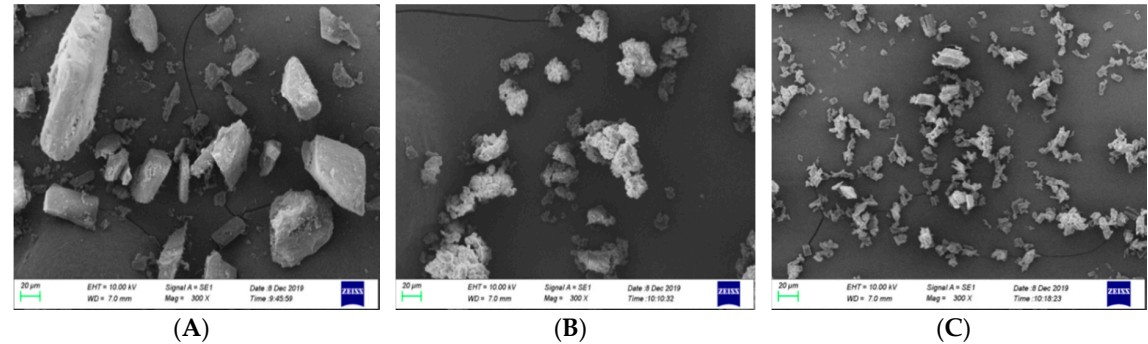

**Figure 7.** Scanning electron microscope images of β-CD (**A**), the physical mixture (**B**), and the LCEO/β-CD inclusion complex (**C**) at 300× magnification.

### 3.8. TGA Analysis

TGA is a means to study the thermodynamics of various samples after thermal degradation happened, and could evaluated the thermal stability of different samples [50]. When the guest molecules of LCEO were embedded into the β-CD cavity, their boiling and melting points could usually transform to other temperatures. The TGA curves of β-CD, physical mixture, LCEO/βCD IC were shown in Figure 8. β-CD has two apparent thermal degradation stages, the first weight loss process occurred at the range of 30~102 °C, which probably attributed to water evaporation, while the second weight loss was located at 311~343.5 °C, which mainly caused by the decomposition of β-CD molecules, such two phases led to a whole weight loss of 80.63%.

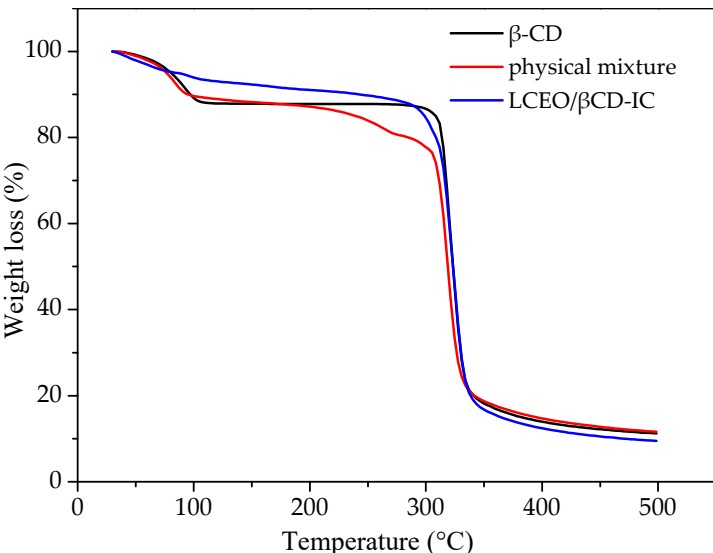

**Figure 8.** Thermogravimetric curves of β-CD, physical mixture and the LCEO/β-CD inclusion complex.

For the β-CD/LCEO physical mixture, the TG curve presented almost similar shape to β-CD, the distribution intervals of thermal degradation slightly differed from β-CD, the initial and second weight losses distributed at the range of 30–93 °C and 292–343 °C, respectively. For the thermal property of LCEO/βCD-IC, it exhibited only one distinct decomposition stage, and was in a state of a relatively slow and stable weight loss prior to the thermal degradation of β-CD. Furthermore, its water loss between 30 °C and 100 °C was weaker than β-CD, indicating that part of crystal water lost in the cavity of β-CD, due to the hydrophobic interaction between LCEO and β-CD. All in all, the weight loss rate of LCEO/βCD-IC was lower than that of physical mixture, noting that the thermal stability of the aimed product was better and desirable.

*3.9. Antifungal Activity*

3.9.1. MIC and MBC Assays

In order to evaluate the antifungal activity of pure LCEO, investigation of MIC and MBC was performed. Table 4 provides the values of MIC and MBC of LCEO sample against three fungi in postharvest citrus. LCEO exhibited antifungal activity against *P. italicum*, *P. digitatum*, and *G. candidum* with MIC values of 3.13 mg mL$^{-1}$, 1.56 mg mL$^{-1}$, and 0.78 mg mL$^{-1}$, respectively. MBCs of LCEO against *P. digitatum* and *G. candidum* were both 1.56 mg mL$^{-1}$. However, the MBC against *P. italicum* was much higher. The differences could be explained by the fungi resistance and sensibility to LCEO.

**Table 4.** Minimum inhibitory concentration (MIC) and minimum bactericidal concentration (MBC), in mg mL$^{-1}$, of LCEO, against three fungi in postharvest citrus.

| Microorganisms | LCEO | |
| --- | --- | --- |
| | MIC | MBC |
| *P. italicum* | 3.13 | 6.25 |
| *P. digitatum* | 1.56 | 1.56 |
| *G. candidum* | 0.78 | 1.56 |

3.9.2. Antifungal Activity Evaluation of LCEO/βCD-IC

LCEO/βCD-IC and pure β-CD were dispersed in agar. The effectiveness of LCEO/βCD-IC, defined as the colony diameter of tested fungi grown in PDA, are depicted in Table 5. When only β-CD was added, the more intense mycelial growth was observed, probably due to the β-CD acting as a

source of nutrient to the mold [46]. However, when LCEO/βCD-IC was diffused in PDA, the mycelial growth of tested fungi was inhibited to some extent. Significant differences can be observed in the colony diameter. It is worth mentioning that the colonial growth of different fungi varies, *G. candidum* obviously germinate faster than *P. italicum* and *P. digitatum*. These results indicated that LCEO/βCD-IC could control the growth of fungi and evidenced a good antifungal activity.

**Table 5.** The colony diameter of tested fungi in potato dextrose agar (PDA) medium containing β-CD and LCEO/βCD-IC.

| Samples | Colony Diameter (mm) | | |
|---|---|---|---|
| | *P. italicum* | *P. digitatum* | *G. candidum* |
| Control | 13.88 ± 0.19b | 15.55 ± 0.12a | 26.90 ± 0.04b |
| β-CD | 15.90 ± 0.12a | 15.61 ± 0.14a | 30.87 ± 0.14a |
| LCEO/βCD-IC | 9.50 ± 0.23c | 9.72 ± 0.02b | 18.65 ± 0.12c |

±Standard deviation ($n = 3$). Mean values with the same letter in each column indicated no significant difference ($p < 0.05$) based on Duncan's multiple range test.

## 4. Discussion

LCEO was an indispensable and favorable antimicrobial to prevent food products from various food-borne pathogens. However, it remains challenging and difficult to make better use of LCEO as a postharvest fungicide. Factors such as light, moisture, air, and temperature all have an impact on the stability of LCEO under nature ambient conditions [21]. Cyclodextrins (CDs) have been recognized as non-toxic wall material, which can form inclusion complex with guest molecules. Previous studies have demonstrated that the complexes of various EOs or their main components with CDs, which have been proven to increase their stability, and contributed to the EOs availability and efficiency applied in fresh fruit and vegetable products as valuable antimicrobials [19,21,40]. In this study, the LCEO/βCD-IC was successfully prepared. The RSM based on BBD was used to determine the best condition of preparation of inclusion complex and obtained the optimum yield. RSM has been widely selected and employed by researchers. Application of RSM based on BBD produced the optimum cinnamon essential oil nanoemulsion [25]. In addition, Maryam found that applying RSM with CD to the edible gelatin film containing EO generated the best physical, mechanical, and antibacterial properties [26]. The EO of holy basil was protected in gelatin, and an optimal encapsulating condition was obtained by RSM and the greatest yield and oil content were realized [24]. These findings suggested that RSM was an empirical technical approach that can be implemented to evaluate the impact of several variables on the response value [51]. In the present work, the results of RSM revealed that the inclusion time of 2 h, temperature of 44 °C, β-CD/LCEO ratio of 4.2 provided the greatest yield of LCEO/βCD-IC. Verification test was also performed to verify the reliability of the model of RSM as previous work reported [36,52]. EE% of the IC at such technique condition was 33.60%, which was nearly closed to the EE% of syringa EO microencapsulation (39.52%) [23], but it was lower than that of thymol inclusion complex (72.30%) [43], which evidenced that the preference of β-CD on EOs depended on the structures of EO molecules.

Furthermore, some physicochemical characterization techniques were employed to demonstrate inclusion complex formation. All indicated that LCEO was successfully included into the host cavity of β-CD. The external surface morphology of inclusion complex could be examined by SEM. LCEO/βCD-IC was clearly equally-distributed compared to physical mixture from the SEM images, which was consistent with the report from Anaya–Castro et al. [53]. Particle size of inclusion complexes was another important parameter. It was hardly reported in previous studies. We found that the particle size of LCEO/βCD-IC was about 17.852 μm, which was smaller than that of microcapsule of holy basil EO in gelatin (392.30 μm) prepared by Sutaphanit et al. [24]. These results suggested that LCEO/βCD-IC optimized by RSM could be evenly distributed, which contributed to the potential application in food preservation field. Moreover, it is significant to note that the thermal stability of

LCEO/βCD-IC was improved after complexation, which could be explained by the fact that LCEO was get adequate protection in the host cavity of β-CD [54]. In addition, FT-IR spectra of LCEO/βCD-IC displayed peaks shifts and changes of intensity when compared to FT-IR spectra of LCEO. However, they showed minor changes when compared to FT-IR spectrum of physical mixtures and pure β-CD. The technique is inadequate to confirm the complexation of LCEO/βCD-IC. According to Andrade [55], several analytical methods are required to characterize the ICs and these methods are complementary to each other. Therefore, XRD have been proposed in our study. It is a simple and rapid crystalline method that allows the verification of complexation of ICs [48]. Our results indicated that the diffraction profile of LCEO/βCD-IC changed a lot and the particles crystallinity was reduced. Moreover, substantial peak broadening was also observed. However, the diffractograms of physical mixture was similar to β-CD, showing no complexation. This trend was consistent with the results of previous studies [56].

LCEO has potential antimicrobial properties [57–59]. According to previous studies, LCEO performed the greatest bacteriostatic effect against *S. maltophilia* [12]. In addition, the antimicrobial effects of LCEO against other pathogens such as cariogenic bacteria and Aspergillus flavus also have been previously reported [5,60]. Although the antibacterial activities of LCEO have been widely evaluated, little attention has been paid to the antifungal properties of it, especially the fungi causing disease in postharvest citrus during storage. In the present study, the antifungal activities of free LCEO were tested against three main fungi in citrus including *P. italicum*, *P. digitatum,* and *G. candidum* and we preliminarily evaluated the antifungal property of LCEO/βCD-IC. All indicated that LCEO and its inclusion complex presented effective antifungal ability.

It is necessary to perform more studies to better understand the antifungal mechanisms of LCEO and its inclusion complex against fungal diseases occurred in postharvest citrus fruit. Additionally, it is also significant to apply LCEO/βCD-IC in more fields such as coating preservation. Currently, the major challenge is to exploit the LCEO/βCD-IC as antifungal additives in food packaging. Furthermore, comparing the differences of antimicrobial activities of unencapsulated and encapsulated LCEO remains a significant challenge.

## 5. Conclusions

In the current study, we prepared the LCEO/β-CD inclusion complex (LCEO/βCD-IC), and optimized it by RSM based on BBD, the greatest yield was obtained at the optimum condition. In addition, the physicochemical characterization demonstrated that LCEO was successfully entrapped into β-CD, forming the inclusion complex. The results of particle size analysis suggested that the particles of LCEO/βCD-IC were smaller and better distributed than physical mixture. Furthermore, thermal stability of the inclusion complex got improved after complexation. In addition, antifungal properties of LCEO/βCD-IC against *P. italicum*, *P. digitatum,* and *G. candidum* frequently occurred in postharvest citrus were tentatively determined. It is promising to develop LCEO/βCD-IC as novel material to reduce the use of chemical fungicides in food industry, so that food safety will accordingly improve a lot.

**Author Contributions:** Y.W. performed the detailed experiments, analyzed the data, drew the plots, concluded the results, and wrote the original manuscript. X.Z. contributed to conceptualization of the research, supervision, and provided significant experimental design. C.Y., X.C., Y.S. and G.L. critically revised the manuscript from format to content. All authors have read and agreed to the published version of the manuscript.

**Funding:** This research was funded by The 13th Five-Year National Key Research and Development Program of China(Grant NO. 2017YFD0401303), the Special Project of Changsha-Zhuzhou-Xiangtan National Independent Innovation Demonstration Zones(Grant NO. 2018XK2006), and Agricultural Science and Technology Innovation Fund project of Hunan Province, China (Grant NO. 2020CX47).

**Acknowledgments:** The author sincerely expresses gratitude to all the friends who helped and encouraged them during this experiment.

**Conflicts of Interest:** There are no conflict of interest to declare.

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
