# Peer review of "β-Cyclodextrin Inclusion Complex Containing Litsea cubeba Essential Oil: Preparation, Optimization, Physicochemical, and Antifungal Characterization"

_coatings, doi:10.3390/coatings10090850_

Round 1
Reviewer 1 Report
Clearly written manuscript, well organized. The scientific research is important and interesting for readers. I recommend accepting your manuscript for publication in Coatings.
Author Response
Thank you for considering our manuscript for publication in Coatings. We are very grateful to you and the other reviewers for the valuable suggestions provided. We will provide more scientific researches in the future. Again, we would like to express our warm thanks to you.
Reviewer 2 Report
The study of essential oil inclusion complexes for food application is an interesting topic.
Attached are some comments and suggestions.

Author Response
Thank you for reviewing our manuscript in your precious time. We feel delighted that our study have attracted your interest. We search some literature as supplement, relevant references were added in Introduction part, the adding part was marked in red. Our research background was more specific. In the further studies, we will pay more attention to the essential oil inclusion complexes, inclusion complexes based on β-CD have great potential in active and smart packaging, We will apply the prepared inclusion complexes to the antimicrobial field for extending shelf life of fruits and vegetables. Again, we would like to express our warm thanks to you.
Reviewer 3 Report
The paper is interesting, the authors deal with the good antimicrobial activities against many pathogens of the Litsea cubeba essential oil (LCEO), but the problems linked with its high volatility and poor water solubility.
I really appreciate the paper, few things are reported in the attached file.
The paper is well written and the all study is well conceptualizated.

Author Response
Point 1: The paper is interesting, the authors deal with the good antimicrobial activities against many pathogens of the Litsea cubeba essential oil (LCEO), but the problems linked with its high volatility and poor water solubility.
Response 1: Thank you for reviewing our manuscript in your precious time. We are really delighted for your appreciation of our manuscript. We initially evaluated the antimicrobial activities of Litsea cubeba essential oil against three common pathogens in citrus fruit, 96-well microdilution broth method for testing MIC and MBC was employed, it was a quick and effective method. Litsea cubeba essential oil is poor water solubility, but it can be well dissolved in ethanol solution. Essential oil in ethanol solution can mixed with PDB broth. Moreover, The microplates were covered with sealing film, in order to prevent LCEO solution from volatilizing, and the results can be observed after incubation at 28 °C for only 48 h, the time does not last too long, which also can decrease the volatility of Litsea cubeba essential oil. Again, we would like to express our warm thanks to you for your valuable comments.
Point 2: I really appreciate the paper, few things are reported in the attached file.
Response 2: We checked the attached file, thanks for your reviewing on our manuscript. We have corrected the errors as you suggested, which were clearly highlighted in red.
Reviewer 4 Report
The manuscript entitled “β-cyclodextrin inclusion complex containing Litsea cubeba essential oil: preparation, optimization, physicochemical and antifungal characterization” develop a LCEO/βCD-IC as novel material to reduce the use of chemical fungicides in food industry. In my opinion, this manuscript is suitable to be accepted in Coatings for Food Technology and System journal after minor revision.
The authors should perform the standardization of international units, since sometimes used ml other mL.
According to section “2.4.1. Single-Factor tests” the preparation times is 1.0, 1.5, 2.0, 2.5, 3.0 h, but in the section 2.42. the preparation time ranged from 1.5 to 2.5, why change? The same happens with temperature and β-CD/LCEO ratios.
The figures resolution should be improve to better visualize.
Line 131: “… ,the…” should be “…., the..”
Line 531: “Moreover, Substantial peak …” should be “Moreover, substantial peak…”
Author Response
Point 1: The authors should perform the standardization of international units, since sometimes used ml other mL.
Response 1: Thanks for your valuable comments and suggestions for us. We carefully checked the manuscript especially the units in it, we standardize the volume unit as mL. We have replaced line 162: 10-4 g.ml-1 as 10-4 g.mL-1, and marked it in red.
Point 2: According to section “2.4.1. Single-Factor tests” the preparation times is 1.0, 1.5, 2.0, 2.5, 3.0 h, but in the section 2.4.2, the preparation time ranged from 1.5 to 2.5, why change? The same happens with temperature and β-CD/LCEO ratios.
Response 2: Firstly, We really thank for your valuable comments and suggestions for us. And we sincerely appreciate your question. Single-factor tests is an experiment in which only one factor changes while the other factors remained unchanged. The specific effects of the factor on the overall experiment can be determined. Single-factor tests can provide an reasonable and suitable data range for the response surface methodology(RSM) experiment.
In our single-factor tests, we presented the preparation time of 1.0, 1.5, 2.0, 2.5, 3.0 h, the results showed that time of 1.5, 2, 2.5 h can provide the relatively better yield of inclusion complex compared with the time of 1.0 h and 3.0 h. So the preparation time ranged from 1.5 to 2.5 h was selected for the further three-factor and three-level RSM experiment, and the optimum preparation condition for LCEO/βCD-IC was precisely investigated. The same as temperature and β-CD/LCEO ratios. Temperature of 20, 30, 40, 50, 60, 70 ℃ was used for the single-factor tests preliminarily, but temperature ranged from 30 to 50 ℃ was selected for the optimization experiment. Similarly, three different β-CD/LCEO ratio of 2 to 6 was more appropriate for the RSM experiment.
Point 3: The figures resolution should be improve to better visualize.
Response 3: Thanks for your sincere suggestion. We realized that the figures in Section 3.1 (Figure 2) is not very clear, we have redraw the plots and the figures resolution have been improved. In addition, other plots were appropriately altered for better visualization.
Point 4: Line 131: “… ,the…” should be “…., the..”
Response 4: Thanks for your tender notice. We have corrected it as you suggested.
Point 5: Line 531: “Moreover, Substantial peak …” should be “Moreover, substantial peak…”
Response 5: Thanks for your kind reminding of us. We ignored the detail of letters. We have corrected it as you suggested.
This manuscript is a resubmission of an earlier submission. The following is a list of the peer review reports and author responses from that submission.